# Pediatric Bronchoscopy for the Adult Interventional Pulmonologist

**DOI:** 10.3390/diagnostics15212769

**Published:** 2025-10-31

**Authors:** Alexa Rangecroft, Alfin Vicencio, Nidhi Kotwal, Siddhartha Dante, Ashutosh Sachdeva, Van Holden

**Affiliations:** 1Section of Interventional Pulmonology, University of Maryland School of Medicine, 665 W Baltimore Street, Baltimore, MD 21201, USA; 2Division of Pediatric Pulmonology, Icahn School of Medicine at Mount Sinai, 1 Gustave L. Levy Place, New York, NY 10029, USA; 3Division of Pulmonology and Allergy, Department of Pediatrics, University of Maryland School of Medicine, 665 W Baltimore Street, Baltimore, MD 21201, USA; 4Division of Pediatric Critical Care, University of Maryland School of Medicine, 665 W Baltimore Street, Baltimore, MD 21201, USA

**Keywords:** pediatric bronchoscopy, pediatric pulmonology

## Abstract

The field of pediatric bronchoscopy is rapidly expanding and enables diagnostic tests and therapeutic maneuvers, benefiting many children suffering from respiratory disease. However, due to a paucity of pediatric providers trained in bronchoscopy, many institutions rely on adult interventionalists collaborating with pediatric care teams to complete these procedures. In this article, we address the adult interventionalist taking on these cases and offer insight into key differences in pediatric anatomy and physiology, unique challenges encountered in this population and explore the equipment available in pediatric sizes. We also consider the future of the field, including broadening pediatric training to enhance capacity to complete these necessary procedures.

## 1. Introduction

The field of pediatric bronchoscopy has rapidly expanded over the past several decades. While rigid bronchoscopy has long been employed by otorhinolaryngologists primarily to retrieve aspirated foreign bodies in children, innovations in flexible bronchoscopy have enabled pediatric pulmonologists to perform diagnostic and therapeutic interventions in the field of respiratory medicine. Despite mechanistic advances in the field, there remain few pediatric specialists trained in interventional bronchoscopy at all centers, thus adult interventionalists often work collaboratively with pediatric care teams to facilitate these procedures. As is commonly said, pediatric patients are not “just small adults”, and adult providers may be unfamiliar with the field of pediatric medicine. In this article, we address adult interventionalists and aim to distinguish anatomic and physiologic differences in the pediatric population, review common indications for bronchoscopy in children, explore available procedural equipment in pediatric sizes, and address additional nuances unique to pediatric patients. Adult providers familiar with these key elements may more confidently, safely, and successfully complete pediatric bronchoscopies. We further explore future directions for the field of pediatric bronchoscopy including enhancing training opportunities and areas of future collaboration.

## 2. Indications for Pediatric Bronchoscopy

Adult bronchoscopy is most often pursued for diagnostic purposes to further assess suspected infection (18.6–42.4%) and malignancy (10.8–32.2%), with fewer studies indicated for evaluation of interstitial lung disease, hemoptysis, and cough [1,2]. Foreign body (FB) aspiration represents 0.26% of bronchoscopy indications in the adult population. In contrast, FB removal remains the primary use case for rigid bronchoscopy in children (32%) [3]. Small rigid bronchoscopy is primarily pursued by otorhinolaryngologists and surgeons and is rarely employed in the field of pediatric pulmonology.

Innovations in the field and manufacturing of smaller scopes have enabled pediatric pulmonary interventionalists to employ flexible bronchoscopy for the evaluation of recurrent respiratory infections (32.2%), stridor (22.6%), congenital airway malformation, and, less frequently, chronic wheezing, hemoptysis, cough, or mediastinal lymphadenopathy [4,5]. While most flexible bronchoscopies remain diagnostic with bronchoalveolar lavage (BAL) applied in roughly 40% of cases, many children benefit from additional therapeutic measures including foreign body removal, cryotherapy, and endobronchial valve placement [6]. Endobronchial ultrasound (EBUS) remains an underutilized tool in pediatrics, though many interventionalists have found success in its use and have used EBUS to mitigate more invasive diagnostic procedures [7].

## 3. Procedural Considerations: Methodology

Though the procedural steps to performing a bronchoscopy in pediatric patients do not differ greatly from those required in adults, there remain significant physical and functional differences between the populations which may impact an interventionalists approach to the procedure.

### 3.1. Size

The most striking difference, unsurprisingly, is the relative size of the patients and, thus, the instruments required. For example, an 8.0 mm endotracheal tube (ETT) is used in most adults and accommodates a 5.8 mm bronchoscope, while pediatric intensivists may use anything from a 3.0 mm ETT in term neonates to 5.0 or 6.0 mm tubes in older children.

Infants using a 3.0 ETT may undergo flexible bronchoscopy with an Olympus H-SteriScope™ Zero (2.2 mm) (Olympus Corporation. 2925 Appling Road, Bartlett, TN 38133, USA), however this scope is limited to visualization as the small diameter does not permit a working channel (WC) for suction or other intervention. Dexterous pediatric interventionalists have used this scope in preterm neonates with 2.5 ETTs, however this poses significant risk of obstruction in the narrow ETT and use should be reserved for experienced pediatric providers. There are many scopes available with working channels to perform various interventions which can be applied in any child using an ETT of size 3.5 or greater. In general, ETTs can safely accommodate scopes with an outer diameter > 0.5 mm less than the inner diameter of the ETT, per manufacturer recommendations (ref: Appendix A). Many institutions recommend 2 mm or greater difference to assure adequate ventilation and oxygenation during procedures.

Given these size-related limitations, laryngeal mask airways (LMAs) remain an excellent alternative to protect the pediatric airway while undergoing bronchoscopy. LMAs are frequently used in out-patient bronchoscopies and those performed on stable, hospitalized patients not requiring prolonged intubation. Additionally, interventionalists may consider extubation to LMA for the purposes of completing a bronchoscopy in more stable, intubated patients. The smallest available LMA, size 1 for neonates weighing less than 5 kg, is compatible with the smallest available single-use bronchoscope, thus enabling diagnostic evaluations and therapies such as suction and BAL to be performed on appropriate neonates (Ref: Appendix A).

### 3.2. Approach

Options for bronchoscopic approach include endotracheal, supraglottic device, transoral, and transnasal. Potential benefits of the transnasal approach include minimal interaction with the gag reflex and the possibility for communication between the proceduralist and the patient, if tolerating anxiolysis or light sedation. This remains an option for older, stable children who may safely complete the procedure under light sedation. Given most pediatric patients require deep sedation, a transnasal approach is often deferred. Additionally, the small size of neonatal and infant nares may further limit instrument options. In cases of tracheomalacia and tracheobronchomalacia, transnasal approach may be indicated to mitigate distortion of the airway [8].

### 3.3. Tools and Procedure

During bronchoscopy, many bronchoscopists perform lavage or take tissue samples from the airway. Options to take on these maneuvers may be limited by airway size and compatible tools.

When performing BAL in the adult population, most proceduralists use 20–60 cc aliquots of normal saline 3–5 times to achieve maximal sample return and safe completion of the procedure. In pediatrics, Ratjen F et al. recommend using aliquots of 1 cc/kg body weight each in patients weighing less than 20 kg and repeating lavage 3 times. For those over 20 kg, 20 mL aliquots may be used [9]. In practice, 30 cc syringes enable many proceduralists to use 30 cc aliquots maximum and they may repeat BAL > 3× to achieve adequate sample size in older children and adults.

Brushes, forceps, and needle aspiration may be employed to collect tissue samples in patients tolerating bronchoscopy with working channels to accommodate these tools. For example, Ultrathin biopsy forceps (Olympus FB-54D-1), small cytology brushes, and needle aspiration may be used with bronchoscopes with working channels ≥ 1.2 mm making biopsy collection possible in procedures involving infants tolerating 3.5 ETT or size 1 LMA. Linear EBUS is limited to those with advanced airways with inner diameters > 8.0 mm, larger than those typically used in pediatrics. However, small radial EBUS devices can be used for those tolerating 5.0 ETT or size 2 LMA. Trans-esophageal EBUS can be considered in patients with airways too small to consider a traditional trans-bronchial approach. Available tools and compatibility are further discussed in Appendix A.

## 4. Procedural Considerations: Anatomy

The pediatric airway differs anatomically from that of adults, with most pronounced differences present in neonates and those less than 1 year old. These differences can pose challenges when intubating or investigating the airway and must be considered prior to bronchoscopy.

### 4.1. Larger Heads

Pediatric heads are proportionally larger than those of adults and classically include large occiputs [10]. This predisposes children to obstructing their upper airways when lying flat. Proceduralists can typically overcome this difference positionally, by including a large shoulder roll and emphasizing extension of the neck.

### 4.2. Limited Pharyngeal Space

Due to large tongues, short mandibles, and frequently enlarged tonsils/adenoids, children are more likely to obstruct their upper airway and have limited space for intervention. Given the hypotonic effects of anesthesia required for intervention, children frequently lose tone in the upper airway upon initiation of a procedure, yielding little time to gain airway access before the onset of profound hypoxemia [11].

### 4.3. Larynx

The pediatric larynx lies more cranially than in adults. While adult vocal cords often sit at a 90-degree angle in the trachea, pediatric cords often lie anterior-inferior and posterior-superior. The more proximal nature and shape of the cords can contribute to peri-intubation trauma to the anterior commissure, especially if the procedure is done emergently or without direct visualization.

### 4.4. Cricoid

At birth, the cricoid ring sits at the C4 level. With normal growth and development, this tissue progresses caudally such that it lands around C5 by age 6 and C6 in adulthood. The cricoid cartilage poses the only calcified ring in the infant airway. It is the narrowest portion of the airway and frequently elliptical in nature, while in adults we expect a more cylindrical structure.

### 4.5. Epiglottis

In infants, the epiglottis is more “U shaped” rather than the flat shape we expect in adults. It can also directly obstruct the glottic opening. These anatomic factors make a straight blade (Miller or Wis-Hipple) rather than a curved Macintosh preferable for intubation until at least 2 years of age.

### 4.6. Increased Compliance

It is important to note that the neonatal and pediatric airway is primary made up of soft cartilage without significant calcification. The hyoid bone is the first to calcify, while tracheal rings and the larynx begin to calcify in the teen years. This increases the relative compliance of the pediatric airway, making it especially susceptible to collapse in the setting of negative pressure or partial obstruction, as may be posed during a bronchoscopy.

### 4.7. Wide Angle of Bifurcation

The tracheal bifurcation angle is wider in children less than 10 years old, around 80-degrees, while the adult bifurcation is typically around 60-degrees. The right and left mainstem bronchi assume equal angles until around age 3 years, at which time the right bronchial angle begins to narrow. This explains why several observational studies have found aspirated foreign bodies in the left main stem in toddlers and why older pediatric and adult patients frequently suffer intubation into the right main stem bronchus [12].

### 4.8. Congenital Malformations

The prevalence of congenital airway malformations is estimated to be 0.2 to 1 in 10,000 live births with most common anomalies including laryngomalacia, vocal cord paralysis, subglottic stenosis, vascular compression of the airway, and tracheomalacia [13]. Many children “out-grow” the impact of these anomalies and they are either not present or not symptomatic once they enter the adult population. Thus, adult interventionalists must be educated on and aware of the possibility of these anomalies when performing bronchoscopies in the pediatric population.

## 5. Procedural Considerations: Physiology

Infants consume oxygen more rapidly than adults with some calculations estimating a rate of 6 mL/kg/min vs. 3 mL/kg/min [14]. This paired with a reduced functional residual capacity increases their risk of rapid desaturation with any decrease in respiratory drive or obstruction. These physiologic factors contribute to a reduction in time that a child can maintain their oxygen saturation while apneic (about 1–2 min) while adults can typically maintain their saturations for 4–5 min while apneic. This time is further limited in cases of significant lung disease or other factors impacting tissue-oxygen requirement.

Additionally, infants produce carbon dioxide more rapidly at a rate of 100–150 mL/kg/min while adults typically produce about 60 mL/kg/min. Pediatric tidal volumes are similar to those in adults (per kg/ideal body weight), therefore adequate ventilation is achieved with a higher respiratory rate [15].

### Mechanical Ventilation

Ventilator management should largely be deferred to primary pediatric intensivist and pulmonary teams, while keeping the above procedural considerations in mind.

Children requiring high-frequency oscillator ventilation are typically too unstable, and in the case of extremely premature neonates, often too small, to consider bronchoscopic intervention, though some case studies have demonstrated success [16]. Jet ventilation is under researched and less frequently used in the pediatric population but has, on rare occasions, been implemented during rigid bronchoscopy procedures [17]. There are no known cases of jet ventilation during flexible bronchoscopy in the pediatric population.

## 6. Complications

In adult populations, the most common complications of bronchoscopy are hypoxemia occurring in ~4.9% of cases and bleeding, occurring in 2.1% [2]. In pediatric procedures, the likelihood of transient desaturation is far higher, with a tertiary center retrospective study noting hypoxemia in 26% of cases [3]. A large study on complications of flexible bronchoscopy by De Blic et al. concluded that the risk of major complications involving oxygen desaturation was associated with younger age (<2 years) and presence of laryngotracheal abnormalities [18]. Bleeding risk is similar to adults, with bleeding noted in ~3% of cases. Post-procedural cough occurs in 14% of patients. Rarely, patients can experience a reaction to anesthesia, mechanical trauma (epistaxis, pneumothorax, hemoptysis), laryngospasm, post-lavage fever, or develop an infection.

## 7. Periprocedural Considerations

### 7.1. Pre-Procedural Fasting

Fasting is required pre-procedurally for all patients to minimize periprocedural aspiration risk. However, pediatric patients, especially neonates, cannot tolerate fasting for prolonged periods (Table 1).

If a child less than 1 year old is made NPO for any duration, they should be started on dextrose-containing maintenance fluids until they demonstrate good oral or gastric intake again. If a child between 1 and 3 years old is made NPO, starting dextrose-containing maintenance fluids is often advised if their NPO status is likely to exceed 8 h. Children over 3 years old are typically not at risk for hypoglycemia in cases of brief fasting; decisions to start dextrose-containing maintenance fluids in this population should be made on a case-by-case basis with special consideration for patients with metabolic disorders [19].

**Table 1 diagnostics-15-02769-t001:** Pre-procedural Fasting Guidelines in Pediatrics by Intake Type [20].

Type of Intake	Minimum Fasting Time (Hours)
Clear liquids (water, oral electrolyte solutions)	2
Breastmilk	4
Infant Formula/Fortified Breastmilk	6
Non-human milk, solids	6

### 7.2. Sedation

While anesthesia and analgesia in the adult population typically target patient comfort, pediatric sedation additionally aims to mitigate patient behaviors which may pose a threat to the safe completion of the procedure. For this reason, deep sedation is recommended when performing pediatric bronchoscopies [21].

Ketamine remains a popular choice in pediatric sedation and may be particularly useful in bronchoscopy due to its bronchodilatory effects. Adverse effects of ketamine including increased airway secretions and emergence delirium can be mitigated with pre-emptive administration of glycopyrrolate and co-administration of low-dose midazolam [22,23].

For out-patient procedures, propofol is often preferred for ease of titration with a randomized controlled trial finding that adding an opioid agonist with propofol sedation reduced peri-procedural cough and decreased recovery time when compared with propofol alone [24]. In neonates, midazolam or chloral hydrate alone are often employed with consistent success in achieving deep procedural sedation with a preferable risk profile [25,26]. As with all bronchoscopies, topical and nebulized lidocaine may be administered for comfort and to mitigate the cough reflex peri-procedurally. Topical lidocaine may be especially useful when employing a transnasal approach. Caution should be used in lidocaine administration with a max dose of 5–7 mg/kg to mitigate risk of seizure and arrythmia.

Most pediatric pulmonologists defer sedation to anesthesia or pediatric intensivists given the above requirement for deep sedation. If an adult provider is performing a procedure on a pediatric patient, sedation and analgesia should be deferred to a pediatric intensivist, pediatric anesthesiologist, or an otherwise designated pediatric sedation team.

### 7.3. Consent and Assent

In the pediatric population, consent must be obtained from parents or otherwise legally appointed healthcare decision makers prior to pursuing bronchoscopy. If an adult provider performs or supervises the procedure, consent should be obtained by the adult provider in partnership with the primary pediatric care team. The primary pediatric care team may better answer questions relating to sedation, NPO status, pre- and post-procedure care, while the adult interventionalist may better explain the intricacies of the procedure itself.

Many hospitals benefit from support teams catering to the developmental and emotional needs of pediatric patients. This may include child life specialists, psychologists, and social work teams. Provider teams are encouraged to include these specialists during consent conversations to better facilitate assent from the child, answer their questions at a developmentally appropriate level, and mitigate the patient’s fears and concerns regarding the procedure.

Additional considerations include hospital malpractice coverage. Ideally, institutional medical-legal teams will establish policies clearly stating malpractice coverage, ethical responsibilities, and delineating roles for provider teams to safely and effectively carry out these procedures. For example, in cases where an adult provider may be leading the procedure, a pediatric provider may be present in the room and listed as an assistant. Additionally, pediatric pulmonary interventionalists may sign annual ongoing professional practice evaluations (OPPE) for adult colleagues certified in pediatric bronchoscopy. Such policies should reflect the needs and resources of the individual institution to best serve their patient population while protecting provider teams. Each institution’s policies may then be reflected in their consent forms for this procedure.

## 8. Conclusions

Pediatric bronchoscopy offers diagnostic insights and therapeutic measures necessary to successfully treat many children with respiratory disease. Due to a paucity of trained pediatric interventionalists, increased recognition of the utility of advanced diagnostic and interventional bronchoscopy in the pediatric population, and introduction of new technology that enables such procedures in young children, adult pulmonary interventionalists are increasingly asked to assist pediatric care teams in the care of their patients. While adult providers manage a diverse array of airway presentations and conditions, the anatomy, physiology, and procedural nuances involved when managing pediatric patients differ greatly from the adult population. Adult providers must take care to educate themselves in these differences, be aware of the scopes and tools available for procedures in pediatric patients and work collaboratively with pediatric care teams to achieve safe and successful procedures.

## 9. Future Directions

While collaboration between adult and pediatric care teams has enabled significant diagnostic and therapeutic advances in the field of pediatric bronchoscopy, there remains significant discomfort for most adult providers in performing procedures in children. This concern is appropriate given adult providers have typically not received extensive training in pediatrics nor in the procedural specifics of performing bronchoscopy in children. There remains ample opportunity to expand training in pediatric bronchoscopy to mitigate these concerns.

Further training in specialized pediatric bronchoscopy interventions is often limited by pediatric patient volume. While adult interventionalists typically see high volumes of patients, the anatomic, physiologic, and procedural differences between the adult and pediatric population described above pose limitations in the transferability of skills. Many pediatric subspecialists including intensivists and anesthesiologists have completed training in complementary fields such that their educational and professional background may further enable their success in achieving competency with bronchoscopy. Adult pulmonary interventionalists have demonstrated competence in bronchoscopy procedures but would benefit from formal training in the specifics of pediatric bronchoscopy prior to pursuing procedures in the pediatric population.

Pediatric interventionalists acknowledge the need to enhance training for motivated individuals to achieve competence in pediatric bronchoscopy. The Pediatric Chapter of the American Association for Bronchoscopy and Interventional Pulmonology (AABIP) has offered specialized tracks at certain conferences including lectures and hands-on simulations to encourage enhancing education in the field.

There remains an opportunity for pediatric interventional bronchoscopy specialists to design a credentialing course geared towards colleagues motivated to achieve competence in pediatric bronchoscopy. The course may benefit from the historic collaboration between adult and pediatric providers in the field, such that students may increase volume of procedures by learning on adult and pediatric patients with an emphasis on pediatric-specific care.

## Data Availability

No new data were created for the completion of this project.

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
