# Peer review of "Pediatric Bronchoscopy for the Adult Interventional Pulmonologist"

_diagnostics, 2025, doi:10.3390/diagnostics15212769_

Round 1

Reviewer 1 Report

Comments and Suggestions for Authors

With pleasure I read your manuscript. It is worth to be read as it summarizes well the anatomical basics for pediatric patients. I would suggest that you include EBUS bronchosopes also in the table and mention the transesophageal access in smaller patients (see study that you also cited).  Minor aspects: Line 54 - why do you mention valve placement? To my knowledge there is not even a published case report for the use in children. And reference 6 should be checked as content does goes along with the reffered part. 

Author Response

1. I would suggest that you include EBUS bronchosopes also in the table and mention the transesophageal access in smaller patients (see study that you also cited). 
Thank you for this insight. EBUS information has been added to the data table in the appendix. I also added a brief description of compatible EBUS in Section 3.3; line 139-143.
I included information regarding transesophageal approach in section 3.3; line 139-143. I did not include this information in the data table in the appendix as the table specifically emphasizes compatibility with ETTs and LMAs.

3. Minor aspects: Line 54 - why do you mention valve placement? To my knowledge there is not even a published case report for the use in children.
Thank you for this question. There have been several successful cases of endobronchial valve placement in children. I have included an appropriate citation now following the mention of valves in Line 54: Yang M, Derespina K, Grant C, Vicencio A, Murthy R, Kaushik S. Bronchopleural fistula management in a pediatric patient requiring extracorporeal membrane oxygenation. Perfusion. 2025 May;40(4):1049-1053. doi: 10.1177/02676591241268367.

4. And reference 6 should be checked as content does goes along with the reffered part. 
Thank you for catching this! The prior reference 6 is now reference 11. The source was used to pull information regarding pediatric airway differences and is now matched with the appropriate data. 

Thank you for your diligent review and thoughtful commentary. 

Reviewer 2 Report

Comments and Suggestions for Authors

Dear Diagnostics Editors:

Thank you for the opportunity to review “Pediatric Bronchoscopy for the Adult interventional Pulmonologist.” This is a review article focused on key bronchoscopy concepts that adult IP physicians need to be aware of when caring for pediatric patients.

This article is well written and quite comprehensive in key anatomical consideration differences between pediatric and adult patients.

  1. Could the authors offer any visuals/diagrams in comparing anatomical differences between pediatric and adult patients?
  2. Could the authors comment on use of small rigid bronchoscopes as well as the use of any special tools that would be used compared with adult rigid bronchoscopy?
  3. Would the use of jet ventilation strategies be helpful for pediatric patients?

Author Response

  1. Could the authors offer any visuals/diagrams in comparing anatomical differences between pediatric and adult patients?
    Thank you for this question. I agree visuals can be very helpful. Such visuals/diagrams exist in previously published manuscripts including those cited in our review. We have no new data to include in an original diagram. Readers can find helpful diagrams in many of our cited resources, including: Harless J, Ramaiah R, Bhananker SM. Pediatric airway management. Int J Crit Illn Inj Sci. 2014 Jan;4(1):65-70. doi: 10.4103/2229-5151.128015.

  2. Could the authors comment on use of small rigid bronchoscopes as well as the use of any special tools that would be used compared with adult rigid bronchoscopy?
    Thank you for this question. In the pediatric practice, small rigid bronchoscopy is typically performed by ENT and Pediatric Surgery, with Pediatric Pulmonology most often deferring the procedure to those specialties. Further, our manuscript specifically focuses on flexible bronchoscopy, with rigid bronchoscopy falling outside the scope of our review.
    Should readers have a similar question, I've addressed this distinction in a new line: Section 2, lines 54-56.

  3. Would the use of jet ventilation strategies be helpful for pediatric patients?
    Thank you for this question. In pediatrics, jet use is anecdotal, largely untested, and extremely rare. There are no cases of flexible bronchoscopy being attempted on a pediatric patient on jet. Jet has been used in case studies in pediatric patients undergoing rigid bronchoscopy, though this procedure is outside the scope of our review.
    To better address this question, I've added a short section: 5.1 Mechanical Ventilation, lines 219-229. This section briefly recommends deferring mechanical ventilation management to primary teams when assisting with bronchoscopy procedures. I also touch on jet and oscillator use in pediatrics and their implications for bronchoscopy. This section includes 2 new citations of case studies to support.

    Thank you for your diligent review of our paper and for your comments!
